# Development of a Multiscenario Planning Approach for Urban Drainage Systems

**Soon Ho Kwon** [1], **Donghwi Jung** [2] and **Joong Hoon Kim** [2,*]

1   Department of Civil, Environmental and Architectural Engineering, Korea University, 145, Anam-ro, Seongbukgu, Seoul 02841, Korea; rnjstnsgh90@gmail.com
2   School of Civil, Environmental and Architectural Engineering, Korea University, 145, Anam-ro, Seongbukgu, Seoul 02841, Korea; donghwiku@gmail.com
*   Correspondence: jaykim@korea.ac.kr; Tel.: +82-2-3290-4722

**Abstract:** A traditional urban drainage system (UDS) planning approach generally considers the most probable future rainfall scenario. However, this single scenario (i.e., scenario-optimal) planning approach is prone to failure under recent climatic conditions, which involve increasing levels of uncertainty. To overcome this limitation, an alternative is to consider multiple scenarios simultaneously. A two-phase multi-scenario-based UDS planning approach was developed. Scenario-optimal solutions were determined for a set of scenarios in Phase I, as the traditional planning approach, while common elements across the scenarios were identified and used to consider components-wise regret cost concept for Phase II optimization, from which a compromise solution was sought. The storm water management model was dynamically linked with the harmony search algorithm for each phase optimization model. The proposed approach was demonstrated in the planning of the grid-type drainage networks of S-city. The compromise solution was compared with the scenario-optimal solutions (Phase I) with respect to cost effectiveness and system performance under scenarios that were not considered in the planning phase.

**Keywords:** urban drainage systems; multi-scenario-based planning; two-phase optimization; pipe installation and size design

## 1. Introduction

Planning robust urban drainage systems (UDSs) to anticipate future climatic conditions is essential for regional communities to mitigate the effects of potential severe flooding. However, accounting for uncertain future conditions in UDS planning is a complex task, and engineers or designers often face difficult decisions.

The traditional approach to UDS planning generally focuses on a single future condition (e.g., the most probable rainfall scenario) that is perceived to be the most likely to occur; this is known as the single-scenario-based optimization approach. However, this approach does not ensure system robustness to satisfy performance requirements under various future conditions. For example, the lowest cost UDS design obtained with a single rainfall scenario with a 50-year return period, although conservative, redundant, and involving high economic investment, would fail in several years when unanticipated future events unfold, for example, transition of a spatial and temporal rainfall pattern of a region and construction of a large apartment complex.

Considerable effort has been devoted to considering scenario(s) in the water resources engineering domain such as the following: evolution of land use and impact assessment [1–10], water distribution system (WDS) design [11,12], watershed management [13,14], water resources management in natural river basins [15], and UDS design [16–19]. The focus of these studies was to obtain the most feasible

solution for the most probable scenario. In previous studies, although solutions for individual scenario(s) have been effectively presented, most of these studies have not provided compromise solutions or integrated policies for scenario-planning-based models.

However, an alternative way to overcome the limitations of these studies is to simultaneously consider a wide range of multiple probable future scenarios, i.e., the multi-scenario planning approach. In the field of WDS, Kang and Lansey [20,21] demonstrated the effect of considering the multi-scenario approach by highlighting the vulnerability of the single-scenario solution under different future scenarios because of the solution's bias to the single scenario considered. Specifically, Kang and Lansey [20] introduced the concept of regret cost (RC), which quantifies over-design and under-design investments of a design solution in different future scenarios. The compromise solution across multiple scenarios was finally obtained by limiting the RC and comparing the optimal solutions of individual scenarios and the compromise solution under different scenarios.

In the water resources engineering domain, one of the research groups in the UDS domain, Ngo et al. [22], introduced a multi-scenario-based design approach, i.e., a two-phase design method in which the individual scenario optimal solution was sought in the first phase and the compromise solution was determined in the second phase, based on the identification of common elements and the calculation of RC. However, the aforementioned study contained several significant limitations which included: (1) RC was calculated at a system-wide level; (2) variation of temporal rainfall distribution was disallowed; and (3) flooding was disallowed (the so-called "fail-safe" design) which hindered verification of the system's hydraulic performance in expelling flood-level volumes of water in the proposed model. In addition, various temporal and spatial distributions of rainfall were not incorporated in the scenarios considered. In summary, a compromise solution from the aforementioned model did not effectively respond or adaptive to recent climatic change conditions, which lead to considerable uncertainties regarding the operation of the UDS.

Multi-scenario-based UDS planning that considers both spatial and temporal rainfall patterns is required to ensure stable UDS performance. The simplified overall system level (Distribution A in Figure 1) in previous studies did not consider individual components in the RC-calculation process. Thus, to compare the compromise and individual solutions, RC calculations must also consider the individual components (e.g., pipe sizes and manhole depth) (Distribution B in Figure 1). Simultaneously accounting for stable UDS performance and the various rainfall patterns would provide critical insight into an effective UDS design and planning for fluctuating future conditions.

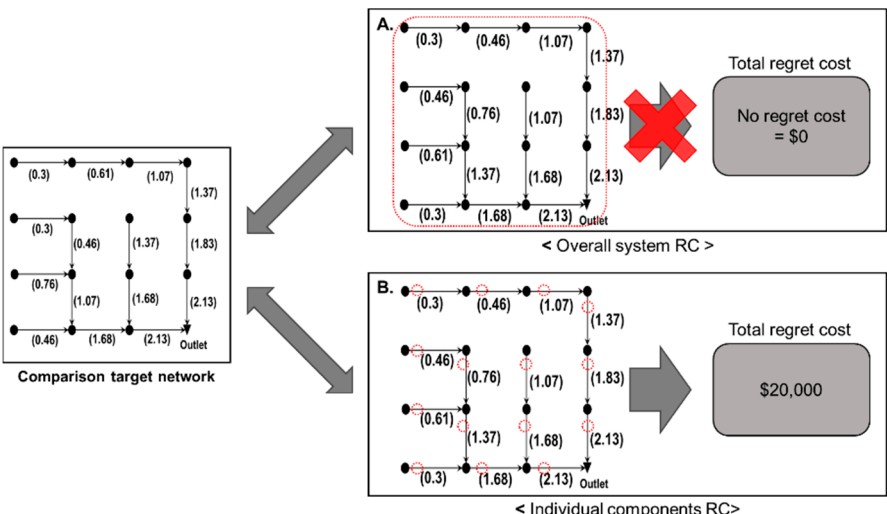

**Figure 1.** Schematic of regret cost (RC) concept at two levels. (**A**) Overall system; (**B**) individual components.

In this study, the pipe design and layout for a given set of future scenarios were optimized. The following future scenarios were developed: (1) different spatial and temporal rainfall patterns and (2) specifications of manholes for drain-inflow distribution in the nodes of the network. These scenarios were applied to obtain an optimal UDS solution to provide guidelines and assist with decision making during the UDS planning. The proposed model consisted of a two-phase optimization process as follows: (1) A scenario-optimal solution was obtained in the scenario-optimal planning model (Phase I) and (2) the common elements across all scenarios were identified, and the total RC was calculated considering individual component-wise RCs to obtain a compromise solution (Phase 2). In the proposed model, to obtain an optimal solution at each phase, a hydraulic–hydrologic simulation linked with a metaheuristic algorithm was implemented for each phase of the optimization process. The proposed model was validated in the UDS planning of a hypothetical grid-type network. Finally, the compromise solution in the proposed approach was compared with those in the scenario-optimal planning model with respect to cost effectiveness and stability of the UDS performance.

## 2. Methodology

The following sections describe the details of the proposed two-phase multi-scenario-based UDS planning model and study network.

### 2.1. Model Overview

The proposed model included four steps to derive a compromise solution that effectively performs across future scenarios, with an acceptable level of investment. In Step 1 (scenario development), multiple future scenarios were produced first by identifying and prioritizing the most uncertain design factors (e.g., unpredictable rainfall pattern from climate change and construction of a large commercial complex) in a planning area and, then, combining each extreme end of the factors (e.g., two different potential construction areas of a large commercial complex). A series of brainstorming sessions was organized in Step 1 with decision makers and related stakeholders. Steps 2 and 3 are the first and second phases of the proposed two-phase multi-scenario planning model, respectively. In Phase I (Step 2), a scenario-optimal planning solution was sought for each individual future scenario to minimize the total system cost. In Phase II (Step 3), the common elements across multiple scenarios were identified by comparing those scenario-optimal solutions, which were used to bound the range of decision variables. Within the decreased solution space, the compromise solution optimization (Step 3) constituted identifying a compromise solution across all the scenarios to minimize the total cost with a constraint on the sum of component-wise RCs, i.e., overpayments or supplementary payments. More details of the regret cost are described later in this section. Finally, in Step 4, the single and multiple solutions, considering both a reasonable planning investment and UDS performance, could be compared and determined by decision makers and other stakeholders. The aforementioned steps of the proposed model are presented in Figure 2.

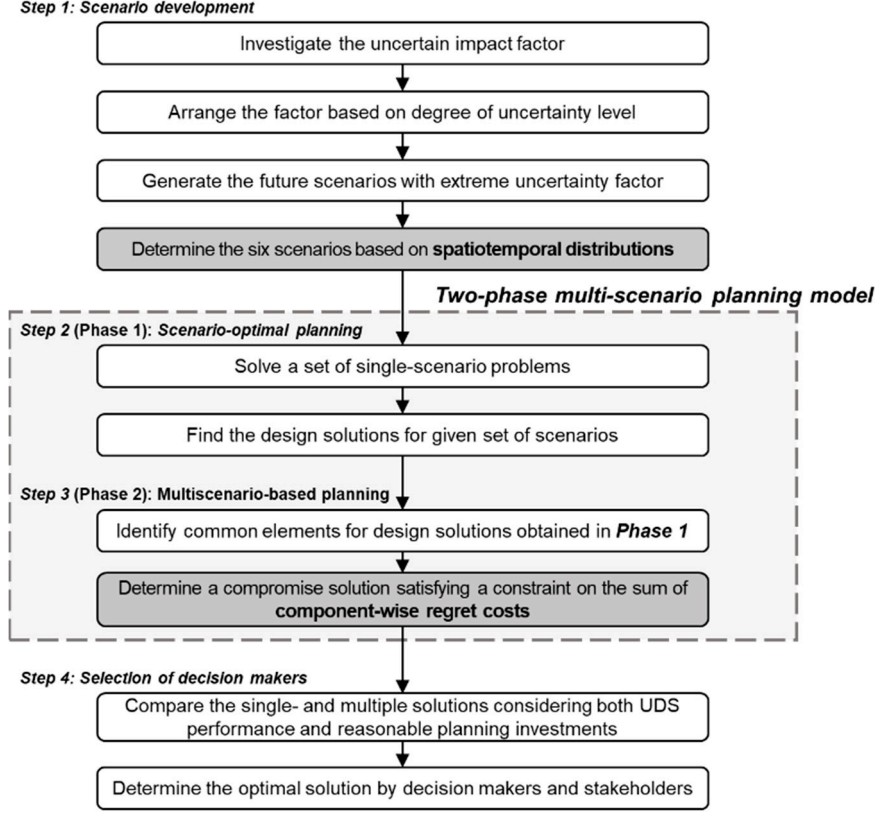

**Figure 2.** Flowchart of the proposed model.

### 2.2. Scenario Development

A UDS planning process should provide a highly flexible compromise solution to prevent severe flooding in any future condition. Given that assumptions and forecasted conditions inevitably change over time, an effective UDS plan should adapt to changing future conditions with minimum investment and impact.

This study adapted a scenario-planning process based on Schwartz' process [23], to construct plausible future scenarios considering various spatiotemporal rainfall patterns. Decision makers and stakeholders first identified the uncertain impact factors in the planning of a UDS of interest (e.g., unpredictable rainfall pattern resulting from climate change and construction of a large commercial complex). A series of brainstorming sessions was organized in this step. Subsequently, these factors were sorted according to their uncertainty level (a factor of greater uncertainty was ranked priori than a factor of lower uncertainty). Each city had a different list and order of uncertain factors because every city faced different developmental and environmental issues. Then, a single scenario was finally constructed by coupling each end of multiple factors. The remainder of this section describes the scenario and development procedures in the study area, S-city.

In this study, it was assumed that S-city planners proposed three city expansion strategies to handle population growth, i.e., expansion of the northwestern corner, midtown, and southeastern corner area in a planning area. This expansion resulted in an increased inflow to nodes in the corresponding regions (Figure 3). Thus, it was critical to consider potential variations in inflow spatial distribution in the scenarios.

As mentioned in the above description, the uncertain factors (i.e., spatial and temporal distribution conditions) identified for this city were the temporal rainfall pattern according to climate change and the spatial inflow variation according to the development of different corners of the city. Two different time distributions of rainfall and three different spatial rainfall distributions were combined to generate a total of six scenarios, as presented in Figure 3. The former time distributions of rainfall were the second and third quartiles of Huff rainfall distributions [24]. The three different spatial distributions of

inflow, including the base 1 q into each node in the system, were (1) additional 1 q to the northwestern corner area (total 2 q at 5 nodes in the area), (2) the same to the midtown area, and (3) that to the southeastern corner area.

In this study, all six scenarios generated were assumed to occur with equal probability because no further information to determine the probability was provided.

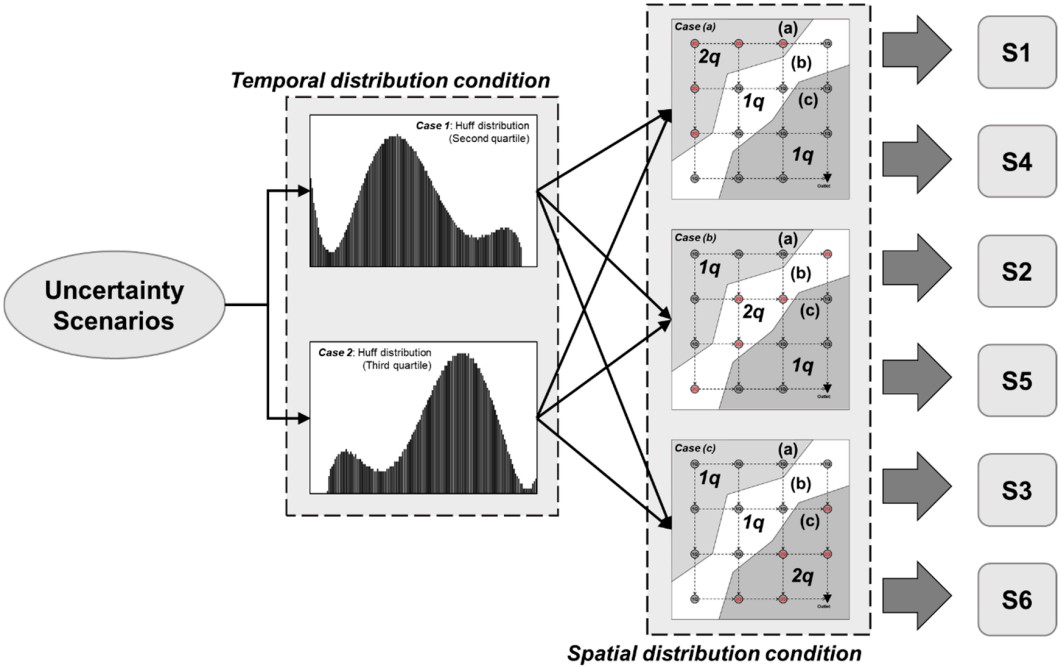

**Figure 3.** Schematic of future scenarios based on uncertainty factors.

### 2.3. Two-Phase Multi-Scenario-Based UDS Planning

The proposed model consisted of a two-phase multi-scenario planning model (Figure 4). Phase I was the scenario-optimal planning solution model to seek the optimal planning solution for each single scenario. In Phase II, the common elements (e.g., pipe sizes and manhole depth) were identified from the scenario-optimal solutions obtained from Phase I. Subsequently, the compromise planning solution that performed well across multiple scenarios was identified in Phase II, by limiting the proposed component-wise RC consisting of the sum of overpayments and supplementary payments. Finally, the UDS performances of the compromise solutions obtained from Phase II, across future scenarios, were confirmed and compared with scenario-optimal solutions from Phase I. The following subsections summarize the formulation of each phase optimization model.

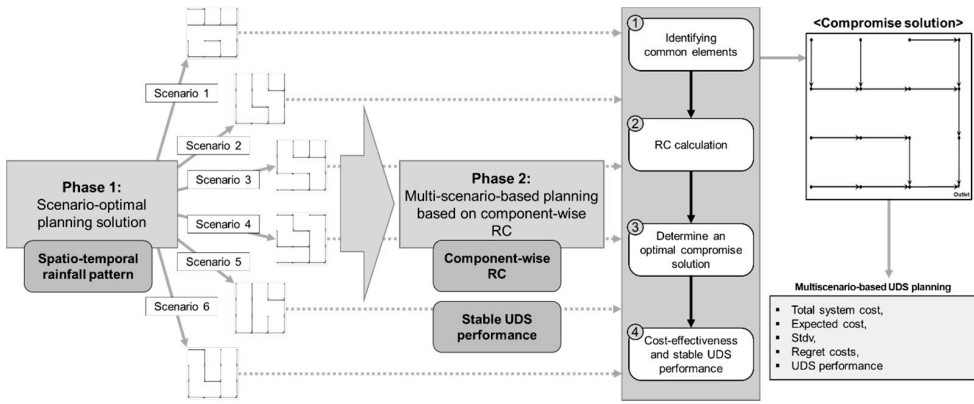

**Figure 4.** Schematic of the proposed two-phase optimization model.

2.3.1. Phase I: Scenario-Optimal Planning Solution

The purpose of Phase I optimization was to simultaneously determine the pipe sizes and layout in order to minimize the total system cost and satisfy a set of predefined constraints. The total system cost is the sum of pipe construction cost (PipeCC), manhole depth cost (ManholeDC), and penalty cost (PenaltyC), as defined by

$$\text{Minimize } F = \text{PipeCC}(X) + \text{ManholeDC}(X) + \text{PenaltyC}\left(X|\omega^i\right) \tag{1}$$

and subject to

$$G\left(X|\omega^i\right) > 0 \tag{2}$$

where $X$ is a set of decision variables (i.e., pipe diameter, layout, and manhole depth); $\omega^i$ is $i$th future scenario, which is a combination of different rainfall patterns and nodal inflow distributions; and $G$ is a set of constraints that can be checked based on the UDS simulation under $\omega^i$. The PipeCC is expressed as

$$\text{PipeCC} = \sum_{j=1}^{N} C_c\left(D_j\right) \times L_j \tag{3}$$

where $C_c$ is the construction cost according to the unit cost of a pipe of a certain diameter per unit length (USD/m), $L_j$ is the length of the $j$th pipe (m), $D_j$ is $j$th pipe's diameter (m), and $N$ is the total number of pipes. The ManholeDC is computed according to

$$\text{ManholeDC} = \sum_{k=1}^{M} M_c \times MD_k \tag{4}$$

where $M_c$ is the unit cost of a manhole (USD/m), $MD_k$ is the depth of $k$th manhole (m), and $M$ is the total number of nodes.

To simultaneously consider UDS stability and the total system cost of the optimal solution, the penalty cost term was appended to the objective function, because the performance of the UDS (e.g., flooding due to UDS) designed by the proposed model must be considered. PenaltyC is a penalty cost that is activated if any of the constraints is not satisfied, but it also includes a function that represents the UDS performance by computing system flooding volume. PenaltyC is formulated as:

$$\text{PenaltyC} = \alpha \cdot \sum_{k=1}^{M} FV_k \tag{5}$$

where $\alpha$ is a penalty constant and $FV_k$ is the flooding volume at node $k$ (m$^3$/s).

The constraints included in $G$ in Equation (2) were (1) pipe-size hierarchy and (2) cover depth. First, the pipe diameters in a UDS should be designed such that the diameter of the downstream pipe ($D_{down}$) is greater than or equal to that of the upstream pipe ($D_{up}$), because it is more likely in practice that a greater volume of water is eventually drained downstream. The constraint for pipe diameters is expressed as follows:

$$D_{up} \leq D_{down} \tag{6}$$

The cover depth constraint limits the vertical Euclidean distance between the ground surface and the top of the pipe, i.e., cover depth. In other words, the cover depth at node $k$ should be greater than or equal to the minimum cover depth allowed ($C_{\min}$):

$$C_k \geq C_{\min} \tag{7}$$

Note that the manhole depth was decided based on cover depth, manhole offset, and pipe diameters.

The scenario-optimal planning model defined by Equations (1) to (7) calculates PenaltyC using the storm water management model (SWMM) [25] linked with the harmony search algorithm (HSA) [26]. The former is a hydraulic–hydrologic simulation model, whereas the latter is a metaheuristic optimization algorithm. The scenario-optimal planning solutions obtained in Phase I, by considering economic investment and system performance with flooding volume, were the best alternative for each individual scenario and together served as the basis when identifying the compromise solution in Phase II that performed well across all multiple scenarios.

2.3.2. Phase II: Multi-Scenario-Based Planning Based on Component-Wise RC

The multi-scenario-based planning model in Phase II was proposed to identify a compromise solution that could adequately withstand a set of scenarios. Phase II included two steps. In Step 1, the scenario-optimal solutions, i.e., $X^* = [X^{1*}, X^{2*}, \dots, X^{i*}]$, obtained from Phase I, were analyzed to identify the common elements. For example, the potential diameter of a pipe that has a range of 0.76–1.22 m in the scenario-optimal solutions was limited in the identified range in seeking the compromise solution (Step 2). As with Phase I, Phase II was also solved using the SWMM linked with the HSA to seek a compromise solution.

In Step 2, the multi-scenario-based planning model determined both the pipe sizes and the layout, to minimize the total system cost constrained by the allowable maximum RC. Note that the total system cost incorporates the penalty cost that is computed considering UDS performance. The proposed model in this study was formulated as:

$$\text{Minimize } F(X|\boldsymbol{\omega}) \tag{8}$$

and subject to

$$G(X|\boldsymbol{\omega}) > 0 \text{ and} \tag{9}$$

$$RC_i \leq RC_{\max} \tag{10}$$

where $F$ and $G$ represent abbreviated forms of Equations (1) and (2), respectively; $\boldsymbol{\omega}$ is a vector of all scenarios; $RC_i$ is the RC in scenario $i$, which represents the sum of overpayments and supplementary payments in the system; and $RC_{\max}$ is the allowable maximum RC. The total RC in scenario $i$ is expressed as

$$RC_i = RC_i^O + RC_i^S \tag{11}$$

where $RC_i^O$ and $RC_i^S$ are the overpayments and supplementary payments in scenario $i$, respectively. In this study, the component-wise (e.g., pipe diameters and manhole depth) RC was computed by comparing the scenario-optimal solutions (Phase I) and the potential compromise solution generated in Phase II. The overpayments and supplementary payments in scenario $i$ are expressed as

$$\begin{cases} RC_i^O = \sum_{n=1}^P \left( RC_{i,n}^O \right) \\ RC_i^S = \sum_{n=1}^P \left| RC_{i,n}^S \right| \end{cases} \tag{12}$$

where $n$ represents the individual components, which can be links or nodes and $P$ is the total number of components.

The overpayment and supplementary payment in scenario $i$ were calculated using the sum of each component $n$. The overpayment and supplementary payment in scenario $i$ for each component $n$, with constrained conditions between the scenario-optimal solutions (Phase I) and the potential compromise solution generated in Phase II, are expressed as

$$\begin{cases} RC_{i,n}^O & \text{if } X_n^{i*} - X_n^{comp} > 0 \\ RC_{i,n}^S & \text{if } X_n^{i*} - X_n^{comp} < 0 \end{cases} \tag{13}$$

where $X^{i*}$ is scenario-optimal solution in scenario $i$, obtained from Phase I and $X^{comp}$ is the potential compromise solution generated by Phase II.

The overpayments and supplementary payments were computed by comparing the $n$th components (links or nodes) of $X^{i*}$ and $X^{comp}$.

### 2.4. Study Network

The simple grid network is a section of a representative flooding area in S-city, South Korea (Figure 5) and was used to develop the proposed model. The discharge and flow of the urban drainage network was determined by gravity only in the study network (i.e., the no-pump condition). The S-city network consisted of 15 nodes, 15 potential links, and one outlet.

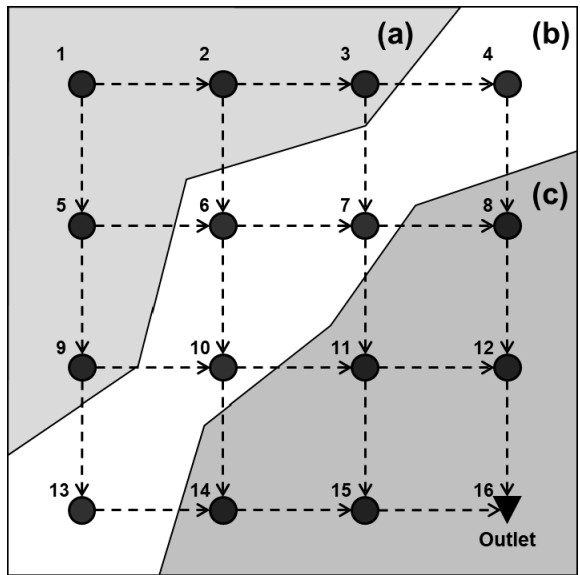

**Figure 5.** S-city urban drainage network. (a) northwestern corner; (b) midtown; and (c) southeastern corner sectors.

This study involved several assumptions and simplifications as follows: (1) the pipe size is selected from among 13 commercial diameters of a pipe (i.e., 0.30, 0.46, 0.61, 0.76, 0.91, 1.07, 1.22, 1.37, 1.68, 1.83, and 2.13 m) or non-pipe case is available (the pipe could not be installed at the 1 to 5 nodes, because of the pipe installed at 1 to 2 nodes); (2) the manhole depth is selected from construction standards (less than 1.5, 1.8, 2.1, 2.4, 3.0, and 3.6 and more than 3.6 m) and a minimum depth more than 0.5 m is available; (3) all pipes are designed from a circular concrete material with a Manning's roughness coefficient of 0.01; (4) the cover depth in the proposed model is more than 0.5 m, because only the freezing depth was considered; (5) the curve number of each subcatchment in the proposed model is set to 95 because most areas in Seoul are paved, with low infiltration. The other design conditions in the proposed model were identical to those in the simple grid network, i.e., the area of each subcatchment was 0.5 ha; the length of each potential link was 220 m; and the surface slope was 0.005.

The upstream-to-downstream node had to be connected with one pipe only (no more than two pipes could be installed by one upstream node). These simplifications and assumptions were considered in the scenario-optimal planning for all scenarios, to ensure a consistent comparison between the results.

A series of rainfall designs was handled by the Huff distribution method [14] and used to determine the time distribution of rainfall. A return period of 30 years was considered for the probability-based design rainfall. This study considered a total of six scenarios within two categories: (1) two different types of time distributions (i.e., second and third quartile generated by the Huff distribution method) and (2) variation of inflow distribution at each node. The inflow to critical nodes due to new

developments in the new urban planning project of S-city was assumed to be 760 L/s (2 q = 2 × 380 L/s). For example, if the city expanded to the northwestern corner, then, the drain inflow at the sector (a) manholes (Case (a)), and the total inflow from urban area (a) would be 10 q (3800 L/s). For the other scenarios (Case (b) and (c)), the inflow to manholes in the mid-town and the southeastern corner increased to 2 q, and the total inflow from (b) and (c) was 10 q (3800 L/s).

## 3. Application Results

To apply multi-scenario planning based on component-wise RC (Phase II), the scenario-optimal planning solution model (Phase I) was first analyzed as a single-scenario problem to identify its limitations and to investigate the RC, cost statistics (cost variance and expected cost), and system performance (flooding).

### 3.1. Scenario-Optimal Planning Solution for Individual Scenarios (Phase I)

In Phase I, the S-city drainage network was independently designed using the scenario-optimal planning model (Equations (1)–(7)) for all future scenarios (i.e., S1 to S6). Table 1 summarizes certain statistics of the optimized dimensions of pipes and manholes and the cost components of the scenario-optimal solutions. The penalty cost was computed using Equation (5), considering the total system flooding volume in the solutions. The six scenario-optimal designs (D1 to D6) were obtained for the corresponding future scenarios (S1 to S6).

**Table 1.** Scenario-optimal designs for multiple future scenarios.

| Planning/Scenario [a] | | D1/S1 | D2/S2 | D3/S3 | D4/S4 | D5/S5 | D6/S6 |
|---|---|---|---|---|---|---|---|
| Pipe | Cost ($M) | 1.466 | 1.462 | 1.318 | 1.372 | 1.823 | 1.357 |
| | Max. size (m) | 2.13 | 1.83 | 1.83 | 1.83 | 2.13 | 1.83 |
| | Min. size (m) | 0.3 | 0.3 | 0.3 | 0.3 | 0.3 | 0.3 |
| Manhole | Cost ($M) | 0.205 | 0.190 | 0.203 | 0.195 | 0.194 | 0.162 |
| | Max. depth (m) | 4.0 | 4.0 | 4.0 | 3.6 | 3.6 | 3.6 |
| | Min. depth (m) | 1.5 | 1.5 | 1.5 | 1.8 | 1.8 | 1.5 |
| Penalty | Cost ($M) | 0.601 | 0.507 | 0.372 | 0.609 | 0.127 | 0.382 |
| System material cost ($M) | | 1.671 | 1.652 | 1.521 | 1.567 | 2.017 | 1.519 |
| Total cost ($M) | | 2.272 | 2.159 | 1.893 | 2.176 | 2.144 | 1.901 |

[a] Scenario-optimal planning (P1 to P6) was the optimal solution corresponding to each scenario (S1 to S6).

While D6 (obtained from S6 in the third quartile of the Huff distribution with a high inflow at the southeastern corner) has the lowest system material cost, the largest investment was required to handle S5 in the same Huff distribution with a high inflow at midtown (D5 was obtained here). The system material cost for D5 was approximately 75% of that of D6. However, regarding the total cost, including the penalty cost (i.e., the system material cost and penalty cost), D3 incurred the lowest cost, whereas D1 was the most expensive. In addition, the total cost for D3 was approximately 83% of that of D1. Thus, the lowest cost D3 scenario-optimal solution leads to potentially small flooding volumes.

Figure 6 shows the hydrograph of the network flooding (system performance) in the scenario-optimal solution under six scenarios (Figure 6a in the second quartile rainfall and Figure 6b in the third). High nodal inflows were transited from north to south in scenarios S1 to S3, as shown in Figure 6a and in S4 to S6, as shown in Figure 6b. Low penalty costs for D3 and D5 were supported by the lowest peaks in the flooding hydrographs in Figure 6a,b, respectively. Therefore, it was confirmed that directing more inflow to the outlet (i.e., development near the outlet, D3) and investing more in pipes and manholes (D5) would result in a less severe failure of the UDS.

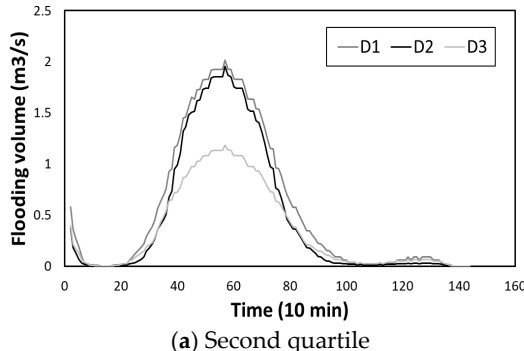 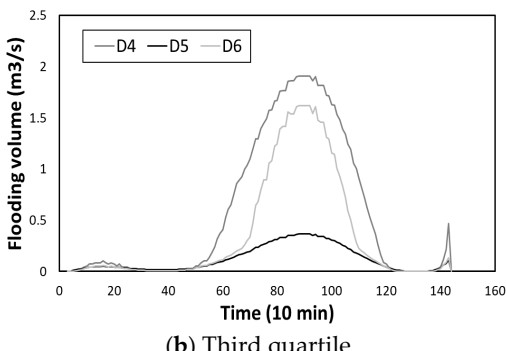

(**a**) Second quartile  (**b**) Third quartile

**Figure 6.** Flooding hydrograph in scenario-optimal solutions under corresponding individual scenarios. (**a**) Second quartile and (**b**) third quartile.

### 3.2. Identification of Common Elements

In the previous section, the scenario-optimal planning solutions (D1 to D6) were obtained from the single scenarios (S1 to S6) to demonstrate their limitations and system performance. However, decision makers or system planners have to select an optimal solution to be implement which should be robust across different future scenarios. This describes the investigation of the common elements, which are the pipe sizes and manhole depth.

Figure 7 shows the ranges of pipe size and manhole depth obtained from the six scenario-optimal planning solutions [Figure 7a,b, respectively]. Note that pipes with greater identifier numbers were located more downstream, closer to the outlet. The range of the component dimensions tends to increase toward the downstream direction, while its center is shifted up as the overall pipe size and manhole depth increase.

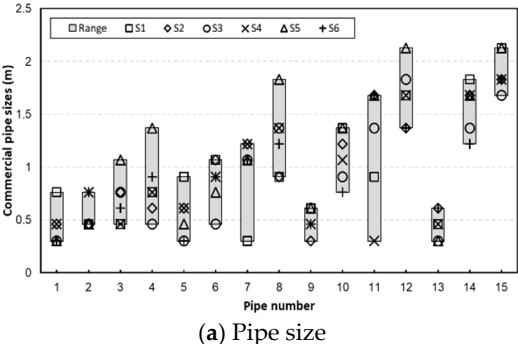 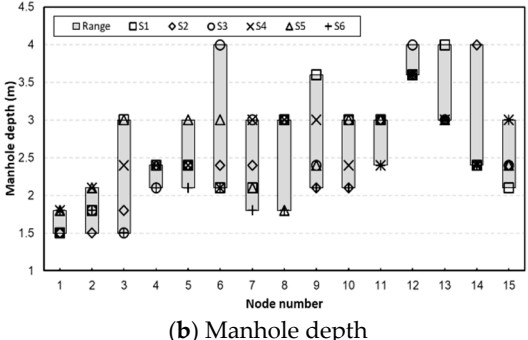

(**a**) Pipe size  (**b**) Manhole depth

**Figure 7.** Optimal component dimensions obtained from the scenario-optimal solutions. (**a**) pipe size and (**b**) manhole depth.

The commercial pipe sizes of Pipes 7 and 11 extend over a wide range, whereas those of the other pipes (3, 4, 5, 6, 8, 12 and 14) extend over a narrow range, for all scenario-optimal solutions. Regarding manhole depth, the majority of nodes in the study network extend over a wide range, whereas the partial nodes, corresponding to Pipes 1, 4, 8, 11, 12, 13, and 14, exhibited extreme dimensions with narrow ranges. These results demonstrate the significant influence of specific uncertainty factors in multiple scenarios.

### 3.3. RC Computation

Table 2 presents and summarizes the system material cost, RC, and the total cost of D1 under six scenarios (S1 to S6). The RCs in Table 2 represent overpayments or supplementary payments that were incurred when unanticipated future events unfolded. For alternative scenarios (S2 to S6), the pipes and manholes determined in D1 should be reconstructed and reinforced at least partially through additional investments (e.g., supplementary payments), or a scenario-optimal solution can be

excessively redundant for certain different scenarios from which the overpayment of RC is computed. D1 from S1 exhibits the highest RC, USD 0.757 million. The penalty costs for S1 and S6 were similar, USD 0.602 and 0.655 million, respectively. The final total cost (optimal + penalty + RC) for S1 and S6 were USD 2.272 and 3.083 million, respectively; their cost difference was USD 0.811 million (approximately 35% more that the final total cost of D1). The range of the total RC for the alternative scenarios (S2 to S6) which is generally indicated high cost, was between USD 0.500 and 0.757 million.

**Table 2.** Cost analysis of D1 determined in Phase I under various alternative scenarios.

| Cost Categories | | S1 | S2 | S3 | S4 | S5 | S6 |
|---|---|---|---|---|---|---|---|
| Pipe construction cost | Optimal | 1.466 | 1.466 | 1.466 | 1.466 | 1.466 | 1.466 |
| | Overpayment | 0.000 | 0.242 | 0.418 | 0.278 | 0.173 | 0.404 |
| | Supplementary payment | 0.000 | 0.238 | 0.270 | 0.184 | 0.530 | 0.296 |
| | Total cost (optimal + RC) | 1.466 | 1.946 | 2.154 | 1.928 | 2.169 | 2.166 |
| Manhole depth cost | Optimal | 0.205 | 0.205 | 0.205 | 0.205 | 0.205 | 0.205 |
| | Overpayment | 0.000 | 0.034 | 0.029 | 0.024 | 0.025 | 0.050 |
| | Supplementary payment | 0.000 | 0.019 | 0.027 | 0.014 | 0.014 | 0.007 |
| | Total cost (optimal + RC) | 0.205 | 0.258 | 0.261 | 0.243 | 0.244 | 0.262 |
| System material costs | Optimal | 1.671 | 1.671 | 1.671 | 1.671 | 1.671 | 1.671 |
| | Overpayment | 0.000 | 0.276 | 0.447 | 0.302 | 0.198 | 0.454 |
| | Supplementary payment | 0.000 | 0.257 | 0.297 | 0.198 | 0.544 | 0.303 |
| | **Total cost (optimal + RC)** | **1.671** | **2.204** | **2.415** | **2.171** | **2.413** | **2.428** |
| **Total RC** | | **0.000** | **0.533** | **0.744** | **0.500** | **0.742** | **0.757** |
| Penalty cost | | 0.602 | 0.624 | 0.655 | 0.632 | 0.612 | 0.655 |
| **Final total cost** | | **2.272** | **2.828** | **3.070** | **2.803** | **3.025** | **3.083** |

Note: All costs are in US Dollars million.

Similar RC analyses were performed for all designs (D1 to D6) to obtain the total cost (optimal + regret costs) for the scenario-optimal planning solutions (D1 to D6) under alternative scenarios; the data are listed in Table 3. The scenario-optimal total costs are in the grey cells in Table 3. The cost statistics include: (1) expected cost, (2) cost standard deviation, and (3) expected RC. The expected cost was estimated as the probability-weighted mean value of the total cost (optimal and RC) for all scenarios, and the standard deviation was calculated as the variability of the total cost under all scenarios. The expected RC is the probability-weighted mean value of the RC (sum of the overpayments and supplementary payments), quantifying the mean cost difference between the optimal and total costs.

**Table 3.** Summary of final total costs including RC and penalty costs, and cost statistics.

| | Scenarios | | | | | | Cost Statistics | | | |
|---|---|---|---|---|---|---|---|---|---|---|
| Designs | S1 | S2 | S3 | S4 | S5 | S6 | E. C [a] | Stdv. [b] | E. RC [c] | E. RC/E. C |
| D1 | **2.272** | 2.828 | 3.070 | 2.803 | 3.025 | 3.083 | 2.847 | 0.280 | 0.655 | 0.230 |
| D2 | 2.726 | **2.159** | 2.658 | 2.584 | 2.824 | 2.487 | 2.573 | 0.213 | 0.494 | 0.192 |
| D3 | 2.418 | 2.345 | **1.893** | 2.220 | 2.462 | 2.361 | 2.283 | 0.190 | 0.558 | 0.244 |
| D4 | 2.570 | 2.548 | 2.678 | **2.176** | 2.873 | 2.689 | 2.589 | 0.212 | 0.526 | 0.203 |
| D5 | 2.953 | 2.870 | 2.782 | 2.873 | **2.144** | 2.831 | 2.742 | 0.272 | 0.662 | 0.241 |
| D6 | 2.711 | 2.216 | 2.340 | 2.384 | 2.601 | **1.901** | 2.359 | 0.263 | 0.542 | 0.230 |

[a] E. C, expected cost; [b] Stdv., cost standard deviation; [c] E. RC, expected RC. Note, costs in bold denote the scenario-optimal cost under the corresponding scenarios.

D1 and D5 exhibited the largest cost variability, which indicates decreasing system performance (greater probability of flooding because of flow congestion) in the alternative scenarios. In contrast, D3 had the highest value for the expected RC from the expected costs as compared with the other designs. Furthermore, D3 had a relatively high expected RC and the smallest cost variability under the alternative scenarios. However, in the event that an alternative scenario occurs, because of the higher expected RC, it would be relatively expensive to expand infrastructures (e.g., increasing pipe sizes, pump stations, and detention reservoir) in D3. Among the six scenario-optimal planning solutions, D1 performed poorly across all cost parameters and had the highest expected cost and RC. In contrast, D2 exhibited lower values for both expected cost and RC than that of the other designs. Thus, D2 could be preferable, because of the moderate cost variability without the other two non-biased cost statistics (i.e., expected cost and RC).

Figure 8 depicts a visualization of the data in Table 3, presenting the cost variability for each scenario-optimal solution under each alternative scenario. The bar plots show the system-material cost, penalty cost (system performance), and RC, which is the sum of the overpayments and supplementary payments.

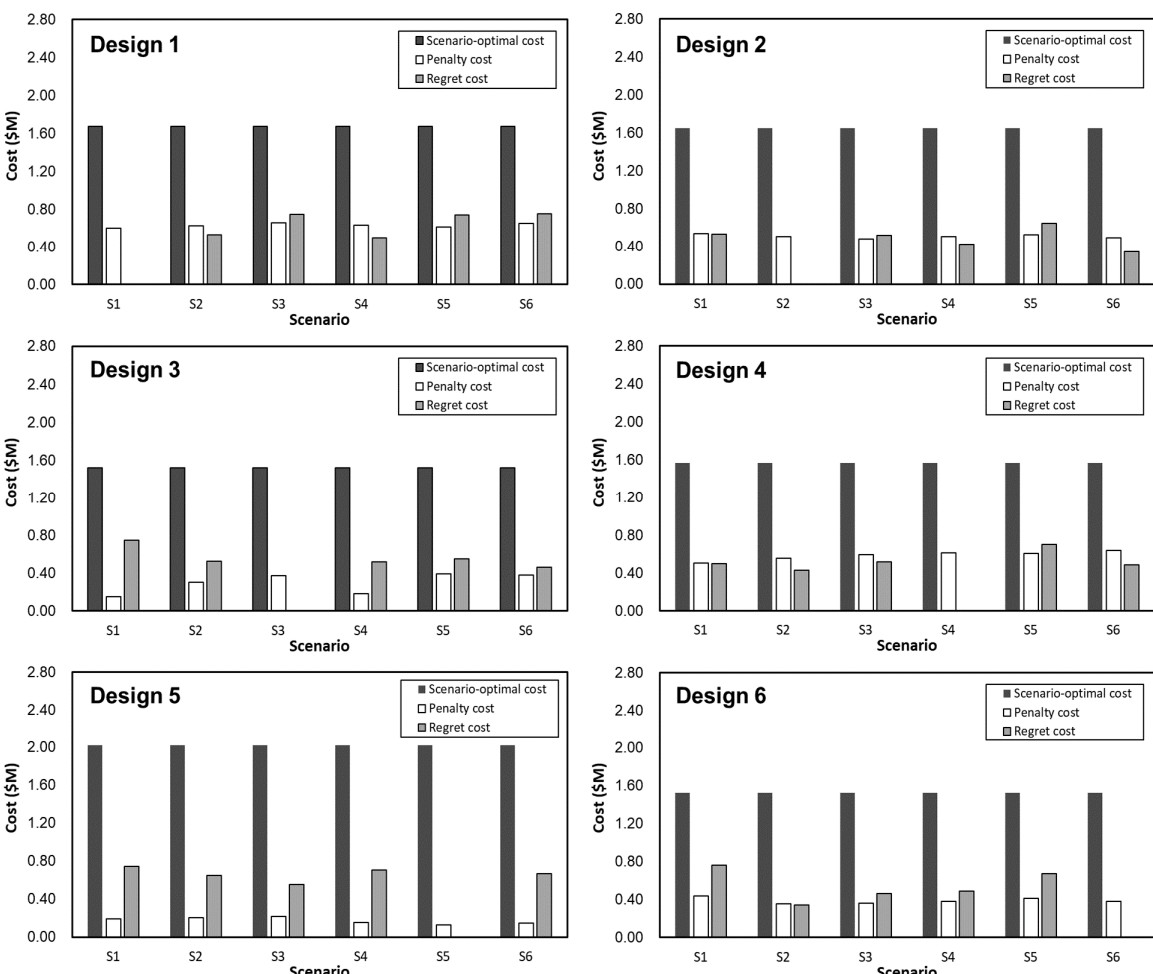

**Figure 8.** Final total costs of each scenario-optimal solution under future alternative scenarios.

The trends of expected cost for D1 and D3 clearly show contrary results (represent Table 3). While D3 had the lowest cost of USD 1.893 million, if S1 occurred, it would incur the highest potential cost, of USD 2.418 million; the difference between these cost values is USD 0.525 million. D1 had the highest cost of USD 2.272 million, and its highest potential cost was USD 3.083 million for S6, the difference

being USD 0.811 million. Generally, the RC obtained from D1 was the result of overpayments, whereas the RC obtained from D3 was due to additional construction costs (i.e., supplementary payments).

In terms of system performance, D1 had the highest penalty cost, because of the greater probability for system flooding. In contrast, D5 had the lowest penalty cost as compared with other designs (D1 to D4 and D6), ranging from USD 0.13 to 0.22 million. We confirmed that systems that achieved high or stable system performance (less flooding events) involved higher scenario-optimal costs than other designs. However, their solutions had a limitation, represented by the overpayments or supplementary payments for alternative scenarios.

### 3.4. Multi-Scenario-Based Planning Solution Based on Component-Wise Regret Cost (Phase II)

This analysis utilized lower constraints than a predefined RC (expected RC obtained from Phase I in Table 3) obtained from scenario-optimal planning solutions. Table 4 shows the RC analysis of the multi-scenario-based planning solution based on component-wise RC.

**Table 4.** RC analysis of the multi-scenario-based planning solutions (Phase II).

| Cost Categories | | S1 | S2 | S3 | S4 | S5 | S6 |
|---|---|---|---|---|---|---|---|
| Pipe construction cost | Overpayment | 0.346 | 0.274 | 0.440 | 0.365 | 0.144 | 0.368 |
| | Supplementary payment | 0.087 | 0.011 | 0.032 | 0.010 | 0.242 | 0.000 |
| | Regret costs | 0.433 | 0.285 | 0.472 | 0.375 | 0.386 | 0.368 |
| Manhole depth cost | Overpayment | 0.009 | 0.018 | 0.010 | 0.006 | 0.012 | 0.034 |
| | Supplementary payment | 0.021 | 0.015 | 0.020 | 0.008 | 0.013 | 0.003 |
| | Regret costs | 0.030 | 0.033 | 0.030 | 0.014 | 0.025 | 0.037 |
| Total regret cost | Overpayment | 0.355 | 0.292 | 0.450 | 0.370 | 0.156 | 0.402 |
| | Supplementary payment | 0.108 | 0.026 | 0.052 | 0.019 | 0.255 | 0.003 |
| | Regret costs | 0.463 | 0.318 | 0.502 | 0.389 | 0.411 | 0.405 |

Note: Units of each cost are million U.S. dollars (US$ million).

RC analyses of the compromise solution obtained in the proposed model revealed decreased values (in terms of the RC) as compared with the scenario-optimal solutions (Phase I). It was confirmed that properly adjusting the pipe sizes and manhole depths based on the common elements from scenario-optimal solutions (Phase I) decreased the RC for all scenarios. D2 had the lowest RC of USD 0.318 million, whereas D3 had the highest RC of 0.502 million USD; the difference between these values is USD 0.184 million. In the compromise solution, supplementary payments showed a more pronounced decreasing trend than overpayments. In addition, in terms of the total RC, a more pronounced decreasing trend was observed as compared with the scenario-optimal solutions obtained from Phase I.

Figure 9 shows a comparison between the scenario-optimal solutions (Phase I) and the compromise solution obtained from the multi-scenario-based planning solution (Phase II) based on component-wise RC. The total system cost included the pipe construction, manhole depth, and penalty costs. The total system cost for the compromise solution in each scenario involved a higher investment than those for the scenario-optimal planning solutions, because the compromise solution considered alternative scenarios to minimize RC while reflecting the common elements. In addition, the total system cost of the compromise solution showed a decreasing trend from S1 to S3. Likewise, a decreasing tendency was observed for the total system cost from S4 to S6. It was confirmed that this tendency was related to the impact of the inflow distribution, that is, the close proximity of the critical nodes (double inflows) to the outlet significantly improved and decreased system performance, along with impact on the total system cost.

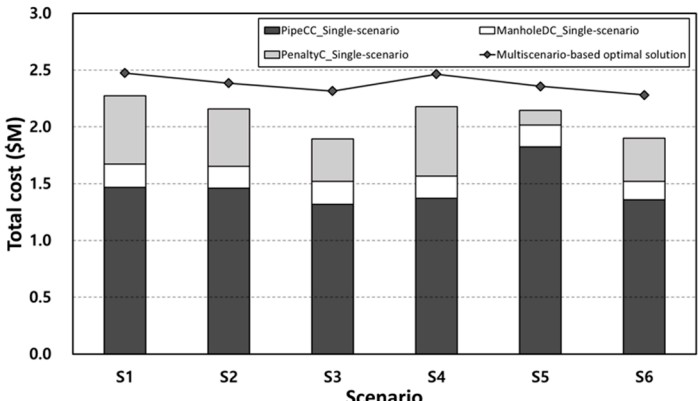

**Figure 9.** Comparison of the final total costs between scenario-optimal solutions and multi-scenario-based compromise solution.

Figure 10 illustrates a comparison of the system performance between the scenario-optimal solutions (Phase I) and the compromise solution obtained from the proposed model (Phase II). Generally, the compromise solution across future scenarios achieved better performance than the scenario-optimal solutions did. However, in S3 and S5, the multi-scenario-based planning solution exhibited a lower performance than the equivalent scenario-optimal planning solution. A compromise solution could exhibit lower performances than scenario-optimal planning solutions because the compromise solution considered all scenarios. Thus, while the scenario-optimal planning solutions obtained from corresponding scenarios could not adequately respond to alternative scenarios without incurring a better system performance, the compromise solution could respond to any alternative scenarios (about 10% improved to system performance), in a cost-effective manner.

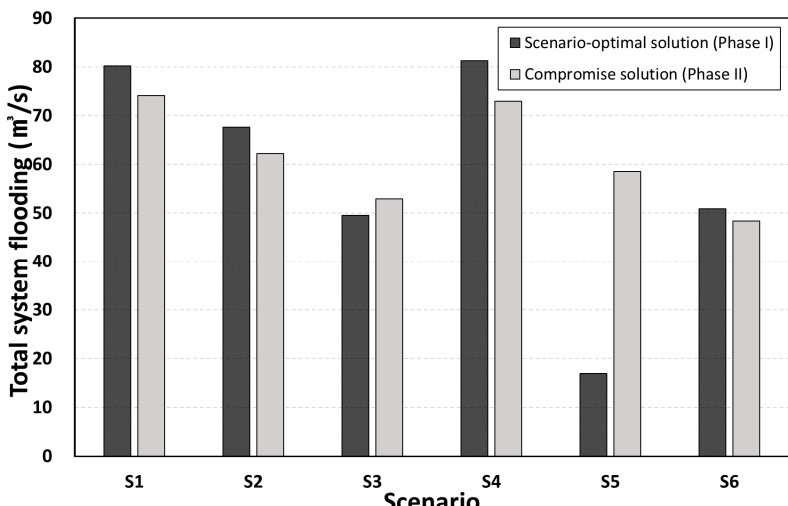

**Figure 10.** Comparison of the system performance between scenario-optimal solutions and multi-scenario-based compromise solution.

## 4. Summary and Conclusions

In this study, the pipe design and layout that minimizes the total system cost for a given set of future scenarios was determined. The future scenarios included the following: (1) different spatial and temporal rainfall patterns and (2) specifications of manholes for drain-inflow distribution in the nodes of the network. The proposed model consists of a two-phase optimization model: (1) the scenario-optimal solution was obtained in the scenario-optimal planning approach (Phase I) and (2) the common elements across all scenarios were identified, and the total RC was computed

considering the individual component-wise RCs to determine a compromise solution (Phase 2). The multi-scenario-based planning model was demonstrated in the UDS planning of a simple grid network, i.e., the S-city drainage network. Finally, the scenario-optimal solution (Phase I) and the compromise solution (Phase II), considering both a reasonable planning investment and system performance, could be presented to decision makers and other stakeholders.

The scenario-optimal planning model (Phase I) delivered a solution that was demonstrated to be cost effective only for its corresponding scenario; for all alternative scenarios, the system performance was poor (high penalty cost) and involved an excessive RC. In addition, in Phase I, the additional requirements for the critical information of the UDS to the inflow distribution, along with other uncertainty factors in future scenarios, were identified. Specifically, the close proximity of critical nodes (doubled inflows) to the outlets significantly enhanced the system performance. Cost analyses for the scenario-optimal planning solutions were also performed; the tradeoff was represented by the expected cost. In addition, the selection of any scenario-optimal planning solution resulted in overpayments and supplementary payments in RC.

To seek a compromise solution, the proposed Phase II optimization model considered critical uncertainties in a multi-scenario-based planning solution approach, by simultaneously including multiple future scenarios. It was confirmed that the proposed model tended to remove the inherent uncertainties in the scenario-optimal planning approach (Phase I). Overall, the results of the compromise solution obtained in Phase II, presented trends of decreasing RC in all future scenarios. In addition, the system performance was considerably improved in the compromise solution as compared with the scenario-optimal solutions (Phase I), although the system material costs were greater than those of the Phase I solutions, because RC was properly decreased while accounting for all future scenarios. Thus, the multi-scenario-based optimization method focused on minimizing the initial construction cost with reasonable adaptability to different scenarios. Generally, the compromise solution across all scenarios achieved better performance than the scenario-optimal planning solutions. Thus, the compromise solution could respond to any alternative scenario and achieve a balance between robustness and flexibility.

This study included certain limitations that could be addressed in future studies. First, the scenario generation, herein, focused only on spatiotemporal rainfall patterns having the same frequency; different design frequencies due to the rainfall patterns were not involved. Second, this study considered the same occurrence probability under different scenarios; an unequal occurrence probability of future scenarios should be considered in future studies. Third, only non-temporal scenarios were considered in this study. Planning or construction with the possibility for modification need to be conducted under various future conditions. Thus, scenario planning considering uncertain factors of a subcatchment (e.g., soil type, land use, slope, and Manning coefficient of the area) or those of a drainage network (e.g., pipe roughness) could be included in long-term strategies, constituting the multi-period planning approach. Finally, this study only included components in the pipe drainage network; future studies on multi-scenario-based planning should consider water infrastructures, such as the capacity of pump stations and detention reservoirs.

**Author Contributions:** Conceptualization, S.H.K. and D.J.; methodology, S.H.K. and D.J.; validation, S.H.K., D.J., and J.H.K.; formal analysis, S.H.K.; investigation, D.J.; writing—original draft preparation, S.H.K. and D.J.; writing—review and editing, S.H.K., D.J., and J.H.K.; visualization, S.H.K.; supervision, J.H.K.; project administration, J.H.K. All authors have read and agreed to the published version of the manuscript.

**Funding:** This research was funded by the Korea Ministry of Environment as "Global Top project" grant number (2016002120004).

**Acknowledgments:** This subject is supported by the Korea Ministry of Environment as "Global Top project (2016002120004)".

**Conflicts of Interest:** The authors declare no conflict of interest.

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
