# Peer review of "Development of a Multiscenario Planning Approach for Urban Drainage Systems"

_applsci, doi:10.3390/app10051834_

Round 1

Reviewer 1 Report

The paper uses multiple futures scenario-based approach to simulate urban drainage and analyze cost-effectiveness. Although it is a well-organized paper with respect to method, there are some theoretical deficits that need addressing.

The literature review is sparse and needs more supporting literature on all areas developed in the paper with a focus on theoretical underpinings. See: Deal, Brian, and V. Pallathucheril. “Developing and Using Scenarios,” in Engaging the Future: Forecasts, Scenarios, Plans, and Projects, ed. L.D. Hopkins, M.A. Zapata  (Cambridge MA: Lincoln Institute for Land Policy, 2007): pp. 221-242. Line 57: “In the UDS domain, Ngo et al. [12] are the only group to have introduced a multi-scenario-based design approach.”Multiple scenario-based simulation approaches are quite common in spatial urban hydrology modelling. The author needs state other (or better) reasons to verify the need of multi-scenario UDS planning. See: Nanía, L. S., León, A. S., & García, M. H., (2015). Hydrologic-hydraulic model for simulating dual drainage and flooding in urban areas: Application to a catchment in the metropolitan area of Chicago. Journal of Hydrologic Engineering Liwanag, F., Mostrales, D. S., Ignacio, M. T. T., & Orejudos, J. N. (2018). Flood modeling using GIS and PCSWMM. Engineering Journal, 22(3), 279–289 Deal, Brian, H. Pan, S. Timm, V. Pallathucheril. “The Role of Multidirectional Temporal Analysis in Scenario Planning Exercises and Planning Support Systems,” Journal of Computers, Environment and Urban Systems  64 (2017): pp. 91-102. Akhter, M. S., & Hewa, G. A. (2016). The use of PCSWMM for assessing the impacts of land use changes on hydrological responses and performance of WSUD in managing the impacts at Myponga catchment, South Australia. Water(Switzerland), 8(11)

Line 13; “It was assumed in this study that S-city planners proposed three city-expansion strategies to handle population growth: expansion of the north-western corner, mid-town, and south-eastern corner area in a planning area” Does S-city refer to a hypothetical city in Korea? Grounds (data sources or existing policies) for setting up the scenarios i unclear. It is hard to catch where the drainage network of S-city (Figure 5) comes from.

Since this research simulates hydrology in “urban area” and looks for spatial approaches, at least land-uses (or land cover) and soil types in the study area should be mentioned. See: Choi, Woonsup, and B. Deal. “Assessing the Hydrological Impact of Potential Land Use Change through Hydrological and Land Use Change Modeling for the Kishwaukee River Basin,” Journal of Environmental Management 88 (2008): pp. 1119-30.

Author Response

Reply to Reviewer 1 (R1)’s comments

We appreciate the reviewer’s helpful comments and have addressed these in detail below. All changes and additions have been highlighted in yellow in the responses and revised manuscript.

R1: 1) The literature review is sparse and needs more supporting literature on all areas developed in the paper with a focus on theoretical underpinnings. Line 57: “In the UDS domain, Ngo et al. [12] are the only group to have introduced a multi-scenario-based design approach.” Multiple scenario-based simulation approaches are quite common in spatial urban hydrology modelling. The author needs state other (or better) reasons to verify the need of multi-scenario UDS planning.

Reply: Thank you for your comment. The main focus of our study is to propose a multi-scenario UDS planning approach that produces a compromise solution, which performs well across multiple future scenarios with minimal overpayment or supplementary cost (i.e., regret cost). Our literature review confirms that the existing UDS design and planning approaches are mostly based on a single and most-probable scenario. Therefore, previous studies were more interested in selecting/considering the single scenario among various alternative future scenarios than using them simultaneously.

We appreciate the reviewer’s suggestion regarding the outstanding literature that contributes to producing and considering multiple scenarios in UDS simulations. However, these studies did not derive a UDS design and planning solution based on multiple scenarios. Our study is targeted at making a contribution with respect to the latter aspect; in particular, multi-scenario optimization techniques are included in our planning approach.

In accordance with this comment, in the revised manuscript, the following references on topics related to land-use or soil type (Line 43, References 1–10) were included.

We also alluded to the study objective in our literature review at Lines 46–49: “In previous studies, although solutions for individual scenario(s) have been well-presented, most of these studies have not provided compromise solutions or integrated policies for scenario planning-based models.”

Additionally, Lines 58–59, have been revised as follows: “In the water-resources-engineering domain, Ngo et al. [22] were one of the research groups in the UDS domain to introduce a multi-scenario-based design approach.”

R1: 2) Line 13; “It was assumed in this study that S-city planners proposed three city-expansion strategies to handle population growth: expansion of the north-western corner, mid-town, and south-eastern corner area in a planning area” Does S-city refer to a hypothetical city in Korea? Grounds (data sources or existing policies) for setting up the scenarios i unclear. It is hard to catch where the drainage network of S-city (Figure 5) comes from.

Reply: In this study, the study network of S-city was a hypothetical one. Additionally, the scenarios considered for the network were constructed based on the rainfall and basin characteristics of Seoul, Korea. The specific value(s) of the basin characteristics in Seoul were included at Lines 256–258 in our revised manuscript, as provided by the Seoul Metropolitan Government (2015).

The following descriptions of future scenarios were established in this study: (1) different temporal rainfall patterns; (2) specification of manholes for drain-inflow distribution in the nodes of the network. As mentioned, conditions, such as the temporal distribution patterns in Seoul, were considered to present considerably different characteristics of heavy rainfall from the S-city network to near urban areas. Thus, when rainfall occurred simultaneously in different urban areas, the temporal distribution patterns in Seoul presented rainfall patterns with different rainfall characteristics between S-city and near urban areas.

Second, we considered the spatial distribution pattern in a set of scenarios. For example, when the population grows or a new city construction project near S-city is scheduled, the urban drainage network will be constructed to correspond with these urban areas (i.e., three areas). If the scale of the new city is large, the constructed urban drainage network will be large in size; in contrast, if the scale of the new city is small, the constructed urban drainage network will be small in size. Thus, S-city’s inflow results exhibited a significant change with respect to population growth or scale of the new city for urbanization.

Future conditions in Seoul were explained by the proposed model in this study set using a total of six scenarios that considered spatial and temporal distribution patterns. This study demonstrated and identified improved cost-effectiveness and system performance, as the results of the multi-scenario-based urban drainage system planning approach (Tables 1–2 and Figures 6–8, respectively).

R1: 3) Since this research simulates hydrology in “urban area” and looks for spatial approaches, at least land-uses (or land cover) and soil types in the study area should be mentioned. See: Choi, Woonsup, and B. Deal. “Assessing the Hydrological Impact of Potential Land Use Change through Hydrological and Land Use Change Modeling for the Kishwaukee River Basin,” Journal of Environmental Management 88 (2008): pp. 1119-30. 

Reply: We thank the reviewer for pointing out the missing information regarding the land-use and soil type. In the revised manuscript, we identified the CN value used for the study network, which embeds the land-use and soil type of sub-basins considered in the rainfall-runoff module SWMM in section ‘2.4 study network’; Lines 256–258: “(5) The curve number of each sub-catchment in the proposed model is set to 95, because most areas in Seoul are paved with low infiltration”; please see the attached file.

In accordance with your comment, we have improved our manuscript by including this vital information regarding the soil type and land-use. Basin information, such as land-use or soil type, has often been considered in the field of urban engineering. However, not only is this information included in the proposed model, which is considered the hydraulic-hydrologic simulation model (i.e., SWMM), but the proposed model is also focused on linking optimization-based modeling with SWMM simulation to develop the multi-scenario planning approach. Thus, this study demonstrates and identifies improved cost-effectiveness and system performance, as results of the multi-scenario-based urban drainage system planning approach.

*Seoul Metropolitan Government (2015) “Report of pump station to improve performance – Sageun-dong in Seoul,” Seoul Metropolitan Government (in Korean).

Additionally, to improve our manuscript based on your comments, we considered and modified the sentence relating to uncertain factors, such as soil type and land-use in sub-catchments in the section ‘Summary and Conclusions’; Lines 456–459: “Thus, scenario planning considering uncertain factors of a sub-catchment (i.e., soil type, land use, slope, Manning coefficient) or drainage network (i.e., pipe roughness) can be included in long-term strategies, constituting the multi-period planning approach”. Furthermore, we intend to develop the scenario planning approach to consider soil or land-use information in further studies.

Finally, as per your comments, the following references were investigated and added to improve our manuscript:

1.          Deal, B. Ecological urban dynamics: the convergence of spatial modelling and sustainability. Building Res. & Info. 2001, 29(5), 381–393.

2.          Deal, B.; Schunk, D. Spatial dynamic modeling and urban land use transformation: a simulation approach to assessing the costs of urban sprawl. Ecological Economics 2004, 51(1–2), 79–95.

3.          Deal, B.; Pallathucheril, V. Developing and Using Scenarios, in Engaging the Future: Forecasts, Scenarios, Plans, and Projects, L.D. Hopkins, M.A. Zapata (Ed., Cambridge MA: Lincoln Institute for Land Policy), 2007, 221–242.

4.          Choi, W.; Deal, B.M. Assessing hydrological impact of potential land use change through hydrological and land use change modeling for the Kishwaukee River basin (USA). J. of Environ. Manag. 2008, 88(4), 1119–1130.

5.          Deal, B.; Pallathucheril, V. Sustainability and urban dynamics: assessing future impacts on ecosystem services. Sustainability 2009, 1(3), 346–362.

6.          Nanía, L.S.; León, A.S.; García, M.H. Hydrologic-hydraulic model for simulating dual drainage and flooding in urban areas: application to a catchment in the metropolitan area of Chicago. J. of Hydro. Eng. 2015, 20(5), 04014071.

7.          Akhter, M.S.; Hewa, G.A. The use of PCSWMM for assessing the impacts of land use changes on hydrological responses and performance of WSUD in managing the impacts at Myponga catchment, South Australia. Water 2016, 8(11), 511.

8.          Deal, B.; Pan, H.; Timm, S.; Pallathucheril, V. The role of multidirectional temporal analysis in scenario planning exercises and planning support systems. Comp., Environ. and Urban Sys. 2017, 64, 91–102.

9.          Deal, B.; Pan, H. Discerning and addressing environmental failures in policy scenarios using planning support system (PSS) technologies. Sustainability 20179(1), 13.

10.       Liwanag, F.; Mostrales, D.S.; Ignacio, M.T.T.; Orejudos, J.N. Flood modeling using GIS and PCSWMM. Eng. J. 2018, 22(3), 279–289.

We appreciate the reviewer’s helpful comments. Please confirm the attached file.

Reviewer 2 Report

The paper is very interesting.

Here are suggested some minor revisions.

Introduction should contains more references. Introduction contains some parts which could be better described in other sections. E.g., the part from row 69 to row 79 is not appropriate in section Introduction, and should be transfered, also with figure 1. Figure 2 needs to be redesigned for a better legibility. Equations need to be resized down, in line of text dimension. The row between 284 - 285 ("Scenario-optimal planning (P1–P6) was the optimal solution corresponding to each scenario (S1–S6)"), and row between 341 - 342 ("E. C: expected cost; bStdv.: cost standard deviation; cE. RC: expected RC.
Note: Costs in bold denote the scenario-optimal cost under the corresponding scenarios") need to be resized down.  The row of Note in Table 4 needs to be resized down. A general revision of Figures dimension.

Author Response

Reply to Reviewer 2 (R2)’s comments

We appreciate the reviewer’s helpful comments and have addressed these in detail below. All changes and additions have been highlighted in yellow in the responses and revised manuscript.

R2: 1) Introduction should contains more references. Introduction contains some parts which could be better described in other sections. E.g., the part from row 69 to row 79 is not appropriate in section Introduction, and should be transferred, also with figure 1. 

Reply: Thank you for your comment. Following your suggestion, in the revised manuscript, we included references regarding topics related to land-use and soil type (Line 43, References 1–10); please see the attached file.

Additionally, as we mentioned in the introduction section, none of the previous studies considered or used either component-wise regret cost for cost-effective urban drainage network design in any multi-scenario planning problems. To the best of our knowledge, this is the first proposal of our methodology, i.e., component-wise regret cost concept in a scenario-planning model, and we believe that it will make a significant contribution to the domain, as agreed by other reviewers.

R2: 2) Figure 2 needs to be redesigned for a better legibility. Equations need to be resized down, in line of text dimension. The row between 284 - 285 ("Scenario-optimal planning (P1–P6) was the optimal solution corresponding to each scenario (S1–S6)"), and row between 341 - 342 ("E. C: expected cost; bStdv.: cost standard deviation; cE. RC: expected RC.

Reply: We modified Figure 2 according to your comment; please see the attached file. Additionally, all equations in our manuscript were resized to the text dimension (i.e., 9 pt); please confirm the attached file.

R2: 3) Note: Costs in bold denote the scenario-optimal cost under the corresponding scenarios") need to be resized down.  The row of Note in Table 4 needs to be resized down. A general revision of Figures dimension. 

Reply: As per your comment, the fonts were resized down for all the Tables in our manuscript, including Table 4. Please see the attached file.

We appreciate the reviewer’s helpful comments. Please confirm the attached file.
